

# Quantifying the impacts of symbiotic interactions between two invasive species: the tawny crazy ant (*Nylanderia fulva*) tending the sorghum aphid (*Melanaphis sorghi*)

Jocelyn R. Holt[1,2], Antonino Malacrinò[3,4] and Raul F. Medina[1]

[1] Entomology, Texas A&M University, College Station, TX, United States of America
[2] Department of BioSciences, Rice University, Houston, TX, United States of America
[3] Institute for Evolution and Biodiversity, Westfälische Wilhelms-Universität Münster, Münster, Germany
[4] Current Affiliation: Department of Agriculture, Universitá degli Studi Mediterranea di Reggio Calabria, Reggio Calabria, Italy

Corresponding author
Jocelyn R. Holt,
Jocelyn.Holt@rice.edu,
holtjocelyn@gmail.com

## ABSTRACT

The establishment of new symbiotic interactions between introduced species may facilitate invasion success. For instance, tawny crazy ant (*Nylanderia fulva* Mayr) is known to be an opportunistic tender of honeydew producing insects and this ants' symbiotic interactions have exacerbated agricultural damage in some invaded regions of the world. The invasive sorghum aphid (*Melanaphis sorghi* Theobald) was first reported as a pest in the continental United States–in Texas and Louisiana–as recent as 2013, and tawny crazy ant (TCA) was reported in Texas in the early 2000s. Although these introductions are relatively recent, TCA workers tend sorghum aphids in field and greenhouse settings. This study quantified the tending duration of TCA workers to sorghum aphids and the impact of TCA tending on aphid biomass. For this study aphids were collected from three different host plant species (i.e., sugarcane, Johnson grass, and sorghum) and clone colonies were established. Sorghum is the main economic crop in which these aphids occur, hence we focused our study on the potential impacts of interactions on sorghum. Quantification of invasive ant-aphid interactions, on either stems or leaves of sorghum plants, were conducted in greenhouse conditions. Our results show that although these two invasive insect species do not have a long coevolutionary history, TCA developed a tending interaction with sorghum aphid, and aphids were observed excreting honeydew after being antennated by TCA workers. Interestingly, this relatively recent symbiotic interaction significantly increased overall aphid biomass for aphids that were positioned on stems and collected from Johnson grass. It is recommended to continue monitoring the interaction between TCA and sorghum aphid in field conditions due to its potential to increase aphid populations and sorghum plant damage.

## INTRODUCTION

When invasive species arrive to a new location, they may establish symbiotic interactions with resident organisms (*Cleland & Mooney, 2001*). New symbiotic interactions have allowed some invasive species to increase population size, expand their geographic range, or disrupt the presence of native species and natural enemies, among other possible scenarios (*Simberloff & Holle, 1999*; *Wilder et al., 2011*; *Traveset & Richardson, 2014*; *Prior et al., 2015*; *Meijer et al., 2016*; *Hulcr et al., 2021*). In particular, mutualisms between native and invasive species or among invasive species may ecologically facilitate biological invasions (*Helms & Vinson, 2003*; *Daane et al., 2007*; *Green et al., 2011*; *Rassati, Marini & Malacrino, 2019*). Ants can adjust foraging preferences to incorporate carbohydrate rich resources for nutrition (*Csata & Dussutour, 2019*), facilitating interactions with honeydew-producing hemipterans. For instance, in field experiments it was reported that invasive red imported fire ant (*Solenopsis invicta* Buren, RIFA) tended invasive mealybugs (*Phenacoccus solenopsis* Tinsley), which increased ant populations, likely due to the provision of honeydew to ants (*Zhou et al., 2012*). In addition, this study found that RIFA was more numerous at this honeydew resource, crowded out the presumed native ants (*Tapinoma melanocephalum* Fabricius), and that mealybug populations increased, potentially due to decreased parasitism in the presence of ants (*Zhou et al., 2012*). While ant-hemipteran interactions can often increase plant damage, ant presence on plants and tending behavior can decrease the presence of other non-hemipteran herbivores and subsequently reduce plant damage (*Styrsky & Eubanks, 2007*). For example, the presence of RIFA on cotton plants was reported to increase cotton aphid numbers while decreasing chewing herbivores (*Diaz, Knutson & Bernal, 2004*). Similarly, in other systems the presence of ants decreased chewing herbivores and increased overall plant health or fruit set (*Styrsky & Eubanks, 2007*; *Singh et al., 2016*).

Invasive ants are often reported to have traits that confer advantages over native competitors, including higher aggression to other ant species, greater reproductive output, a more general diet, or better ability to tolerate disturbance (*Porter & Savignano, 1990*; *LeBrun, Abbott & Gilbert, 2013*; *Calcaterra, Cabrera & Briano, 2016*). Beneficial symbiotic interactions or mutualisms can also promote invasive species growth and establishment (*Traveset & Richardson, 2014*). The interaction between ants (Hymenoptera:Formicidae) and aphids (Hemiptera:Aphididae) is an excellent system to explore potential mutualisms, due to these taxonomic groups' variability in symbiotic interactions over evolutionary history (*Stadler & Dixon, 2005*; *Pringle, 2021*). Although there are many beneficial instances of ants tending aphids, the level of ant attendance towards different aphid species varies (*Stadler & Dixon, 2001*; *Yao, 2014*); with some aphids rarely being tended by ants while others rely on ant tending for increased survival *via* reduction in natural enemies or removal of excess honeydew. In addition, there are instances in which ants may eat aphids, regulate aphid population size, or reduce dispersal (Offenberg, 2001). In contrast to symbiotic interactions among co-evolved organisms (*e.g.*, leafcutter ants and their symbiotic microbes, Lycaenid butterflies and ants, ambrosia beetles and their fungi, etc.,) (*Mueller et al., 2005*), invasive species may engage in new symbiotic interactions in new
locations. The potential for invasive species symbiotic interactions to enhance invasion processes requires further investigation (*Ness & Bronstein, 2004*; *Holway et al., 2002*).

To further understand how a potential mutualism may influence two recently invasive species, we quantified the symbiotic interaction between sorghum aphid (*Melanaphis sorghi* Theobald), which are distributed across the continental US (*Nibouche et al., 2021*) and tawny crazy ant (*Nylanderia fulva* Mayr), which occupy some portions of the southern US. While there are instances of different ant and aphid species engaging in symbiotic interactions across evolutionary history, the interaction of these invasive insect species is relatively recent. Tawny crazy ant (TCA) are known to be opportunistic tenders of a variety of hemipterans, and sorghum aphid are tended by a variety of ant species in Texas (*Wright, 2021*; *Holt, 2022*). Multiple ant species including RIFA, acrobat ants (*Crematogaster* spp. Lund), and TCA have been observed tending sorghum aphids (*Wright, 2021*); Holt personal observations). While sorghum aphids have already resulted in significant economic damage to grain sorghum crops (*Villanueva et al., 2014*; *Zapata et al., 2016*), a potential mutualism between ants and these invasive aphids could increase pest population sizes, potentially resulting in greater damages to sorghum through reduced availability of nutrients, yield reduction, buildup of honeydew which results in sooty mold growth and reduced photosynthesis, and potential pathogen transmission (*Zapata et al., 2016*; *Zapata et al., 2018*). TCA opportunistic tending behavior was recently reported to cause greater plant damage in sugarcane crops (*Pazmino-Palomino, Mendoza & Brito-Vera, 2020*). In Florida, TCA was reported as a pest around the 1990s, and has opportunistically tended both native (*e.g.*, willow aphids, juniper aphids, psyllids, cottony maple scale, kermes scale) and invasive (*e.g.*, citrus whitefly, cowpea aphids, mealybugs) hemipterans (*Sharma, Oi & Buss, 2013*). Both TCA and sorghum aphid have continued to expand their geographic range (*MacGown, 2015*; *Nibouche et al., 2021*), and now these insects occur in many of the same geographic regions.

In addition, we examined whether aphid-host plant adaptation or, more broadly, local adaptation, influenced ant-aphid interactions or aphid biomass. In some instances, host plants may play a role in attracting ant tenders. For example, black bean aphids (*Aphis fabae* Scopoli) preferred genotypes of tansy host plants (*Tanacetum vulgare* L.) that emitted chemical volatile blends including eucalyptol and terpineol, and black garden ants (*Lasius niger* L.) were attracted to these volatile blends, which facilitated ant-aphid interactions by bringing ants in contact with aphids (*Zytynska et al., 2019*). Local adaptation may also influence ant tending rates. Such was the case with cowpea aphid (*Aphis craccivora* Koch), where aphids collected from one location were tended more frequently by ants than those collected from a different location further away (*Katayama et al., 2013*). Thus, aphids and ants adapted to different geographic locations or host plants may generate different mutualistic interactions. Aphids from different host plants were used in our study since sorghum aphids commonly occur on grain sorghum and Johnson grass. More recently sorghum aphid was reported on sugarcane (*Nibouche et al., 2021*), potentially due to aphids blowing over onto this host plant.

Given the opportunistic tending behavior of TCA, we hypothesized that ants would readily tend sorghum aphid in a greenhouse setting. We also hypothesized that this
symbiotic interaction would result in increased aphid biomass, suggesting that in some instances this may be a mutualistic interaction. In order for tending to occur ants need to discover aphids. We hypothesized that aphids located lower on the sorghum stem would be more easily discovered by foraging ants and receive greater tending activity than those located higher up on leaves. To accomplish this, we quantified the tending behavior duration of TCA workers to sorghum aphids. In addition, we analyzed whether aphid biomass was influenced by ant presence or the host plant from which an aphid was collected (*i.e.,* sorghum, Johnson grass, sugarcane). Overall, our study found that symbiotic interactions can establish relatively quickly between recently introduced invasive species, and that this interaction should continue to be monitored.

## METHODS

### Insect collections and rearing

Sorghum aphid (*M. sorghi*) was collected from the leaves/stems of grain sorghum, Johnson grass, and sugarcane on Texas A&M Property and through the Texas Ecological Laboratory. Some previously established lab colonies of aphids were used in these experiments. A total of eight sorghum aphid clone lines were reared for this study (sugarcane = 2, sorghum = 2, Johnson grass = 4; Table S1). Although aphids were collected from different host plant species, all aphids were reared on the grain sorghum variety DELKALB® DKS44-20 (Bayer, St. Louis, MO, USA), which is a tillering hybrid in the medium maturing group that can be grown in the Central Texas Region (*Balota et al., 2013*). Aphid clone lines were generated using single parthenogenetic females to ensure genetic homogeneity within each colony and each colony was kept separate inside fine mesh cages with grain sorghum plants. Rearing room conditions for aphids were approximately 25 to 28 °C, with a relative humidity of 40 to 60%, and a 12:12 Light:Dark cycle. Grain sorghum plants grown from seeds were reared in the greenhouse in 6-inch round pots (McConkey, Sumner, WA, USA) filled with Jolly Gardener Pro Line C/25 soil (Jolly Gardener, Atlanta, GA, USA) and allowed to grow for 2 to 4 weeks before they were used in experiments. All plants used for rearing aphids were of similar size (>20 cm tall).

The tawny crazy ant (*N. fulva*) workers used in these experiments were collected from field sites in Texas (30°43′39.0″N 96°19′26.0″W and the surrounding area in Bryan; since the completion of these experiments, these collection areas now have housing developments on them). A total of eight colonies were collected from Bryan. Population genetic structure analyses of TCA used in these experiments showed that they were genetically similar (*Holt, 2022*). Ant colonies were maintained in a five-gallon bucket filled with the soil surrounding each colony at the time of collection. The inside rim of the bucket was lined with corn starch to prevent ants escaping, which followed a slightly modified version of Drees rearing protocol (*Drees, 2012*) and ants were dripped out following modifications to McDonald's protocol (*McDonald, 2012*). Modifications to Drees protocol included: lining buckets with corn starch instead of talcum powder, foregoing the soapy dishwater moat surrounding the base of the collection bucket, using an artificial nectar solution instead of honey, and excluding the provision of artificial diet. Modifications to McDonald's protocol included:

filling buckets to $\frac{3}{4}$ or more, gently stacking cardboard squares on a stiff wire bridge rather than gluing the cardboard together, excluding the use of a weight in the bucket, and using a five gallon plastic carboy to dispense water rather than a sink faucet. Buckets with ants were placed in a rearing room at approximately 25 to 28 °C, 40 to 60% humidity, and a 12:12 Light:Dark cycle. Each bucket was supplied weekly with a whole cricket and with two vials, one containing reverse osmosis water and the other one containing an artificial nectar solution (100 g sucrose, 90 g glucose, and 53 g fructose dissolved in 1000 mL of reverse osmosis water modified from *Lanza (1991)*; vials were replenished as needed. Ants were dripped at a rate of approximately one drop per second and aspirated out of collection buckets for the experiment. Each collection bucket was split into two or more sub-colonies that included 500 ant workers and one queen. Tawny crazy ant worker numbers are based upon those used in *Katayama et al. (2013)*. Any remaining ants were left in the original collection colony bucket for further studies. One day before the experiment, artificial nectar was removed from all ant colonies to be used (*Katayama et al., 2013*). This was done to promote foraging behavior for honeydew while also reducing ant mortality from starvation. A new water vial and cricket were provided to ants for the experiment.

### Ant-Aphid interaction experimental design

Plant cages built to enclose each experimental unit were placed inside a Fluon-coated (Insect-a-Slip Barrier by BioQuip, Rancho Dominguez, CA, USA) container (Sterlite 58 Quart Storage Box dimensions 59.69 L × 42.88 W × 31.11 H cm). Plant cages were built using PVC pipe (1.27 cm diameter) assembled into a rectangular shape (dimensions 25.4 L × 43.18 W × 76.2 H cm), and mesh bags composed of 100% polyester white voile (86.36 cm L × 182.88 cm H) were used to enclose the PVC frame. As an additional barrier to prevent ants from escaping, the table legs of the greenhouse benches were placed into a container filled with unscented mineral oil (Equate Mineral Oil, Al Ahmandi, Kuwait).

A total of 20 individual aphids from a clone line were transferred onto individual sorghum plants with a damp paintbrush. Ten aphids were placed on the stem and ten were placed on the 3rd leaf from the bottom of each sorghum plant used in experiments. The same aphid clone was used for a pair of experimental units (*i.e.,* a sorghum plant with aphids and ants and a sorghum plant with aphids and without ants). Aphids used in the experiments were from eight clonal lines: four clones from Johnson grass, two from sorghum, and two from sugarcane (Table S1). Each aphid clone was used at least once for an observation period (Table S1).

In the case where aphids were placed with ants, aphids were allowed to acclimate between two to four hours before adding an ant sub-colony (500 workers + 1 queen). Each new ant sub-colony was placed inside a 1-gallon bucket (Lowes, Mooresville, NC, USA) filled approximately 90% with soil (Jolly Gardener Pro Line C/25, Atlanta, GA, USA) centered around a 6-inch round pot (McConkey, Sumner, WA, USA) that contained a plant with 20 aphids. The pot nested inside the bucket was placed inside of a PVC frame with a mesh cage that was tied at the top, and then this was placed inside a large plastic container coated with Fluon at the top rim (Fig. 1). This experiment included aphids collected from sugarcane ($n = 6$), from grain sorghum ($n = 3$), and from Johnson grass ($n = 4$), which were in the

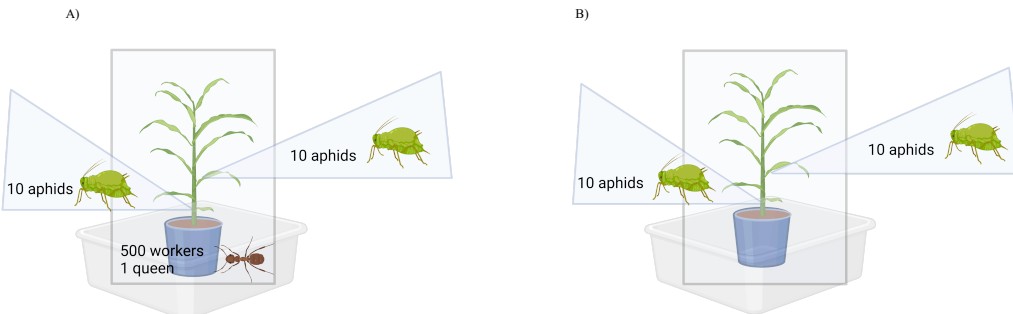

**Figure 1 Observation assembly for ant-aphid interaction experiments.** When experimental units were not under observation, the mesh bag was raised and tied closed. When experimental units were under observation, the mesh bag was lowered. (A) Experimental units with ants consisted of a sorghum plant with 20 reproductive female aphids, 500 ant workers and one ant queen. (B) Experimental units without ants consisted of a sorghum plant with 20 reproductive female aphids only. Created with BioRender.com.

presence or absence of ants ($n = 2$), for a total of 26 experimental units. Within each unit, aphids were either placed on a grain sorghum plant stem or leaf ($n = 2$), yielding a total of $n = 52$ observations ($n = 26$ with ants and $n = 26$ without ants).

## Ant tending observations

After the experimental units were assembled (Fig. 1), aphids were allowed to acclimate for the remainder of the day. The following day, observations were conducted between 8:00 AM and 2:00 PM. Tending behavior observations for each experimental unit were conducted for four consecutive days (except for a pair of pots, which were observed only for two days), and observations were done sometime between the months of May and October of each year. During observations, the mesh cage was lowered for the ant-aphid unit and its corresponding unit with aphids only. Each sorghum plant with ants and aphids was quantified for the duration of total tending behavior (*i.e.,* ants touching aphids either *via* antennating, carrying, drinking honeydew, or walking on them) by all ants over 10 min (based on (*Katayama et al., 2013*)) on either the leaf or the stem. The average amount of time ants tended aphids was taken for the consecutive observations of each sample. The order in which experimental units were observed was randomized each day. After the fifth day, sorghum plants were cut at the base, placed in a large, zippered plastic bag, and stored in the freezer. Aphids on these plants were later collected and used to measure aphid biomass.

## Measuring aphid biomass

Sorghum plants were removed from the freezer and aphids were collected from the observation area using a paint brush. Aphids were placed into a labeled glass vial with acetone. Aphid biomass was measured by removing aphids from acetone filled vials with a paint brush then placing them into a pre-weighed aluminum weigh boat (approximately 1.27 cm diameter by 0.635 cm deep) and allowing them to dry out. Once aphids dried, the weigh boats with aphids were placed onto a precision balance (Mettler Toledo UMX2;

Metter Toledo, Columbus, OH, USA). Weighed aphids (mg) were returned to their vials and kept as vouchers.

### Data analysis of TCA tending and aphid biomass

Data analysis was performed using R $v.$ 4.1.0 (*R Core Team, 2021*) and the packages *lme4* (*Bates et al., 2015*) and *car* (*Fox & Weisberg, 2019*). The package dbplyr/dplyr was used, for data processing (*Wickham et al., 2018*). First, we tested whether tending duration (the time that ants were in contact with aphids, observed within 10 min and averaged over four days) was different from zero using a $t$-test separately for aphids feeding on plant stems and aphids feeding on leaves. Then, we tested whether ant tending frequency was influenced by host plant or aphid position by fitting a linear mixed effect model (LMM) using Type II Wald Chi-square tests to analyze host plant aphids were collected from (*i.e.,* sugarcane, sorghum, or Johnson grass), aphid position (*i.e.,* stem or leaf), and their interaction as fixed factors, and observation year as a random factor; the contrast of leaf to stem used the Kenward-Roger Degrees of Freedom method. We also tested whether aphid biomass was influenced by the factors host plant, aphid position or by the presence of TCAs by fitting a LMM using Type II Wald Chi-square tests and examining all two- and three-ways interactions as fixed factors, and observation year as a random factor. All pairwise comparisons were tested using the packages *emmeans* (*Lenth et al., 2021*) and *multcompView* (*Graves, Piepho & Selzer, 2019*), FDR-correcting $p$ values for multiple comparisons.

## RESULTS

### Sorghum aphid honeydew excretion behavior, ant tending, and aphid biomass

Sorghum aphids generate copious amounts of honeydew, which was observed being flicked away from the insect when it was not tended by ants (https://youtu.be/VJcEpk5jXJw). However, after being antennated by TCA workers, sorghum aphids were observed modifying this behavior to excrete honeydew for ant consumption (https://youtu.be/H8jD5YkBkpM).

We analyzed whether ant tending duration of aphids was significantly different from no tending events. When tending was analyzed by aphid position on plants using a $t$-test, we found that the tending duration was significantly higher than zero when aphids were feeding on stems ($t = 4.68$, $df = 12$, $p = 0.001$, $n = 13$) and leaves ($t = 2.97$, $df = 12$, $p = 0.012$, $n = 13$). After fitting a LMM, we found that only aphid position on plant significantly influenced the TCA tending duration ($\chi^2 = 10.400$, $df = 1$, $p = 0.001$, $n = 26$), with a higher value when aphids were located on stems (estimate $= -3.29$, $p = 0.014$, Fig. 2A), and neither host plant origin ($\chi^2 = 1.905$, $df = 2$, $p = 0.551$), or the interaction between host plant origin and aphid position ($\chi^2 = 1.337$, $df = 2$, $p = 0.513$), influenced TCA workers tending duration towards sorghum aphids.

We also analyzed whether aphid biomass was affected by the host plant aphids were collected from, aphid position, or the presence of TCA using a LMM. We found a significant effect driven by the interaction between host plant aphids originated from, aphid position,

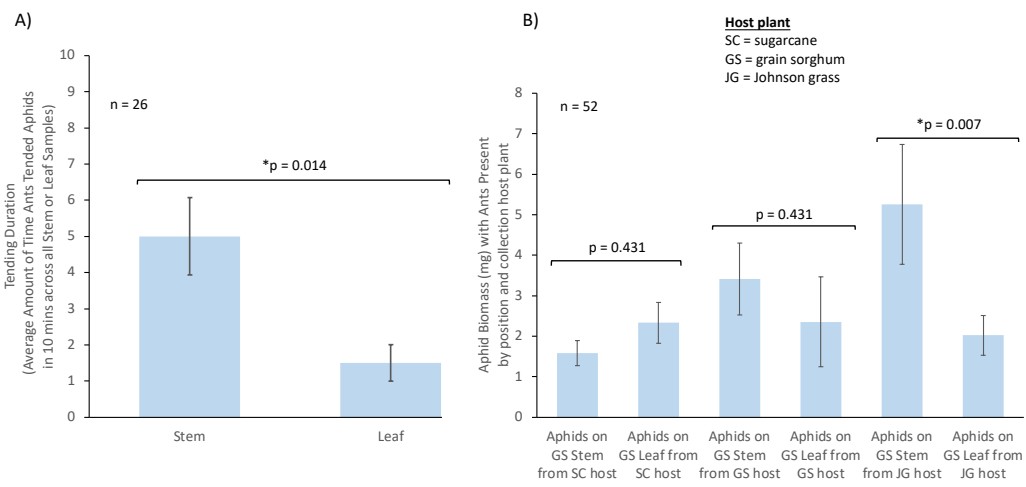

**Figure 2 The influence of aphid position on ant tending duration and aphid biomass.** Where aphids are positioned on a plant can influence ant tending and aphid biomass. (A) Influence of aphid position on plant (stem or leaf) on TCA tending duration ($n = 26$). (B) Influence of aphid position on plant and aphid host plant origin on aphid biomass in the presence of ants (SC = sugarcane; JG = Johnson grass; GS = grain sorghum; SC stem $n = 6$, SC leaf $n = 6$, JG stem $n = 4$, JG leaf $n = 4$, GS stem $n = 3$, GS leaf $n = 3$). Bars represent mean with SE. * represents a significant difference between treatments.

and ant presence ($\chi^2 = 6.93$, $df = 2$, $p = 0.031$, $n = 52$, Fig. 2B). *Post-hoc* contrasts showed that aphids had a higher biomass when feeding on sorghum plant stems compared to those feeding on sorghum leaves, when aphids originated from Johnson grass, and when ants were present in the system (estimate $= -3.243$, $p = 0.007$; Fig. 2B). No differences in biomass were observed for aphids in the presence of ants when aphids originated from sorghum (estimate $= -1.057$, $p = 0.431$) or originated from sugarcane (estimate $= 0.747$, $p = 0.431$), nor in the absence of ants for aphids from sorghum (estimate $= 0.733$, $p = 0.584$), sugarcane (estimate $= -1.860$, $p = 0.053$), or Johnson grass (estimate $= -1.825$, $p = 0.119$).

## DISCUSSION

We found that although tawny crazy ant and sorghum aphid are relatively recent invasive pests in the US and are not previously reported to have coevolved with each other, both organisms engaged in a potential mutualism. This invasive ant antennated sorghum aphids for honeydew while sorghum aphids often modified their behavior from flicking honeydew to secreting honeydew after being antennated by TCA. Similar to other reports of TCA being opportunistic tenders of other hemipterans (*Sharma, Oi & Buss, 2013*), this was also the case with sorghum aphids.

In this study, the position of aphids on a sorghum plant was found to influence TCA worker tending rate. Higher tending occurred when aphids were located on the stem of a sorghum plant than on the leaves. This difference in tending could have been influenced by ants scouting and finding aphids more often when they were located on the stem compared with when aphids were higher up on the leaf. This could also reflect the quality of the

honeydew resource when aphids were located on stems *versus* leaves or the proximity of the resource to the ant nest. While sorghum aphids in field patches of Johnson grass and sorghum were tended by ants on leaves of different heights (personal observation) this was potentially due to increased foraging by larger ant colonies, that honeydew flicked by aphids may attract ants, and/or that aphids walked to different parts of the plant or were moved by ants to facilitate tending. Interestingly, aphids collected from Johnson grass showed increased biomass when positioned on stems *versus* leaves, while this was not the case for aphids collected from sugarcane or sorghum. In addition, this increase in biomass was recorded only when ants were present. It is possible that ants preferentially tended aphids on stems due to this honeydew containing a greater sugar and nutrient concentration than that exuded by aphids on leaves. For instance, soybean aphid (*Aphis glycines* Matsumura) feeding on stems were found to increase in numbers more than those feeding on leaves, likely due to the greater content of amino acids and sugars when compared with phloem contents of leaves (*Nalam et al., 2021*). Since all aphids were reared on sorghum plants in our study, it is possible that the aphids collected from Johnson grass had greater reproductive outputs in the presence of ants due to the more extended availability of Johnson grass throughout the year and increased likelihood that aphids are tended by ants. This is in contrast with aphids collected from sorghum or sugarcane, crops that are grown for a shorter time, which may proportionally reduce aphid interactions with ants.

Understanding the symbiotic interactions of invasive species can aid with pest management efforts. For example, after the invasive mealybug (*P. solenopsis*) arrived in China, it established a new interaction with the native ghost ant (*T. melanocephalum*) in less than five years (*Feng et al., 2015*). This symbiotic interaction benefited ghost ants that received mealybug honeydew, while mealybugs gained ant protection against native parasitoid wasps (*Feng et al., 2015*), increasing population growth of both species (*Zhou et al., 2014*). As a result, invasive mealybugs were likely to cause greater injury (*e.g.*, loss of plant nutrients, leaf chlorosis, decreased seed set/yield, pathogen transmission) to cotton and hibiscus plants in the future. Similarly, the spread of invasive Argentine ant (*Linepithema humile* Mayr) across the US was likely facilitated by the consumption of honeydew produced by native membracids (*Harvey & Wheeler, 2015*). Despite the recent formation of this symbiotic interaction between the aforementioned insects, this tending interaction resulted in damage to both urban and wildland systems (*Menke, Ward & Holway, 2018*). Symbiotic interactions, many of which are beneficial to invasive insects, can therefore facilitate ecosystem disruption and, as a consequence, can result in future potential invasions.

In the scenario of TCA workers and sorghum aphids, a symbiotic interaction may be beneficial to both these insect pests and increase numbers in sorghum fields. In our experiment, visual inspection of TCA colonies at the end of the experiment revealed the production of brood, suggesting a trajectory of colony growth in the presence of sorghum aphids that requires further investigation. This beneficial dynamic has been reported for TCA in other scenarios. In Colombia, opportunistic tending of mealybug (*Antonina* sp.) by TCA resulted in damaged grasslands, which reduced the forage quality and value for

cattle (*Zenner De Polania, 1990*). Similarly, higher sugarcane aphid (*Melanaphis sacchari* Zehntner) presence on sugarcane was reported in Ecuador, where large TCA populations tended these aphids (*Pazmino-Palomino, Mendoza & Brito-Vera, 2020*). Previously, aphid injury towards sugarcane was considered low and aphid population sizes were controlled by parasitoid wasps and other natural enemies. However, TCA tending reduced natural enemy presence (*Pazmino-Palomino, Mendoza & Brito-Vera, 2020*), and resulted in increased sugarcane aphid populations. In these instances, interactions between two invasive pests ecologically facilitated aphid fitness and reduced crop health. Currently in the US, the composition of sorghum aphid natural enemies and ability to control aphids has been reported (*Hewlett, Szczepaniec & Eubanks, 2019*; *Maxson et al., 2019*; *Elliott et al., 2021*; *Faris, Brewer & Elliott, 2022*; *Faris, Elliott & Brewer, 2022*) along with the finding that RIFA did not negatively impact the presence of natural enemies that parasitize sorghum aphids feeding on sorghum (*Wright, 2021*). However, additional assessments of ant-sorghum aphid dynamics is needed, which could include identifying other ant species that are likely to increase aphid biomass or assessing the impact of ant presence on natural enemies in different ecosystems. Potential factors influencing the symbiotic interaction between invasive TCA and sorghum aphid are complex. For instance, conditions reflective of climate change (*i.e.,* increased $CO_2$ and increased temperature) were reported to cause increased honeydew production by aphids (*Kremer et al., 2018*; *Blanchard et al., 2019*), which could result in a greater attraction of ants and reduction in plant nutrients. Factors such as honeydew concentration or chemical composition (*Wright, 2021*), or honeydew bacterial composition (Holt et al., in preparation) could also influence parasitoid or ant attractiveness to honeydew exudates. The intensity of ant-aphid interactions therefore may vary by aphid population, host plant consumption, or geographic location.

In our study, ant tending was observed both at the initial placement of twenty aphids and as aphid numbers increased; although this tending only significantly increased sorghum aphid biomass when those aphids were collected from Johnson grass. This is similar to the finding that sorghum aphid population growth can be variable. In one study, sorghum aphid populations tended by RIFA were reported to have significant population increases when aphid numbers were low compared with when aphid numbers were high (*Wright, 2021*). The number of aphids on a plant may differ from the overall biomass of the aphids (*i.e.,* small aphids *versus* large aphids), which is likely to influence the amount of honeydew that is secreted, and the potential attractiveness of honeydew to ant tenders. Further investigation into how aphid numbers affect ant tending or the long-term impact ant tending has on aphid biomass on a host plant is needed to understand ant and sorghum aphid dynamics. The impacts of sorghum aphid honeydew on the presence of ants and/or on colony growth still need to be quantified. For instance, invasive RIFA were reported to have increased colony growth in the presence of cotton aphid (*Aphis gossypii* Glover) feeding on cotton plants (*Wilder et al., 2011*). An increase in TCA colony growth from consuming honeydew could result in further displacement of native arthropod species or an exacerbation of invasive hemipteran damage.

## CONCLUSIONS

While our study finds that the presence of TCA can increase sorghum aphid biomass in greenhouse settings, there remain unanswered questions about the potential ecological impacts of this interaction. Since the symbiotic interaction between these two insects is recent, with overlapping distributions and potential interactions as early as 2013 (*Meyers & Gold, 2008*; *MacGown, 2015*), it is possible that the intensity of this interaction will increase over time. Symbiotic interactions between these two invasive insects could cause population increases for aphids as well as ants. Focusing on agro-ecosystems, it is still unknown what the potential long-term consequences of ants tending sorghum aphids are on aphid biomass over a growing season or to plant health and grain production. When we shift to natural ecosystems or wildland/green space areas we should consider what roles this symbiotic interaction may play in increasing invasive ant population growth, excluding native insects, or facilitating further invasive organisms. These symbiotic interactions may alter the effectiveness of management practices such as biological control or further disrupt ecosystems. Overall we recommend continued monitoring of the interaction between sorghum aphids and invasive ants.

## ACKNOWLEDGEMENTS

We thank the Texas Ecological Research Laboratory for access for collecting aphids. We also thank our undergraduate researchers for their assistance in maintaining aphid and ant colonies and in making collection vouchers: Jose Torres and Robert Chapa. Thank you to our colleagues who contributed aphids for these experiments: Danielle Sekula, Scott Armstrong, Blain Viator, Greg Wilson, and David Kerns.

### Funding

This work was supported by the National Institute of Food and Agriculture, U.S. Department of Agriculture Hatch Program (TEX09185 to Raul F. Medina). Jocelyn R. Holt received funding from the Texas Ecological Laboratory to conduct this project. This article was written while Jocelyn R. Holt was funded by an AFRI Competitive Grant from USDA NIFA (TEX09837). There was no additional external funding received for this study. The funders had no role in study design, data collection and analysis, decision to publish, or preparation of the manuscript.

### Grant Disclosures

The following grant information was disclosed by the authors:
The National Institute of Food and Agriculture.
U.S. Department of Agriculture Hatch Program: TEX09185.
The Texas Ecological Laboratory.
An AFRI Competitive Grant from USDA NIFA: TEX09837.

## Competing Interests

The authors declare there are no competing interests.

## Author Contributions

- Jocelyn R. Holt conceived and designed the experiments, performed the experiments, analyzed the data, prepared figures and/or tables, authored or reviewed drafts of the article, and approved the final draft.
- Antonino Malacrinò conceived and designed the experiments, analyzed the data, authored or reviewed drafts of the article, and approved the final draft.
- Raul F. Medina conceived and designed the experiments, authored or reviewed drafts of the article, and approved the final draft.

## Field Study Permissions

The following information was supplied relating to field study approvals (i.e., approving body and any reference numbers):

The Texas Ecological Laboratory approved access to private property in Texas to collect aphids.

## Data Availability

The script and raw data are available in the Supplemental Files.

## Supplemental Information

Supplemental information for this article can be found online at http://dx.doi.org/10.7717/peerj.14448#supplemental-information.

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
