# Peer review of "Quantifying the impacts of symbiotic interactions between two invasive species: the tawny crazy ant (Nylanderia fulva) tending the sorghum aphid (Melanaphis sorghi)"

_PeerJ, doi:10.7717/peerj.14448_

## Round 0.1 · original submission · Major Revisions

Dear Authors,
Please submit your point to point revisions as recommended by our reviewer team. An early response would be highly appreciated.

Thank you so much.


Reviewer 1 ·

Basic reporting

This is an interesting study and the authors have collected a very interesting novel finding about the interaction between the two invasive species i.e. aphid and ants. The authors have used cutting edge methodology to conduct this study. Overall the manuscript is well written and structured, however, in my opinion there are very little shortcomings which need to be addressed and are mention in the attached review report

Experimental design

Study experiment is well design and structure and also executed properly. But there are some questions that I have raised and are mention in the attached review report.

Validity of the findings

All the findings of this study are valid and study plan is executed properly. A separate annotative review report is attached

Additional comments

Review report is attached

Annotated reviews are not available for download in order to protect the identity of reviewers who chose to remain anonymous.

Reviewer 2 ·

Basic reporting

-The overall structure of the article is consistent with what is expected of a professional research article.

-The abstract and introduction sell the importance of understanding interactions between invasive species. The language is mostly clear, though at times could be more specific. For instance, the writing could be more specific about what is poorly understood about these interactions and immediately describe how the present study addresses these needs.

-Some background information regarding the evolution of trophobiotic interactions between ants and honeydew-producing insects is needed. I agree that it is critical to understand the interactions between these two species, but the presentation of the interaction of the ant and aphid species studied in this work does not take into account the deep evolutionary history (tens of millions of years, as seen in amber fossils) that ants as a group have as generalist tenders of honeydew-producing insects.

-The results are relevant to the hypotheses.

Units are unclear in Figure 2. This figure should be interpretable as a standalone from the rest of the manuscript. The authors should define the initialisms: "SC" "JG" and "GS". I recommend adding symbols or letters indicating statistical significance between groups.

-I recommend inclusion of a table in the results section summarizing the statistics.

Experimental design

-The study appears to be relevant to the aims and scope of PeerJ.

-The hypotheses are clearly stated and are testable.

-Importance of understanding relationships between invasive species is made clear, though more specific information as to how this study addresses the need could be presented more immediately in the manuscript.

-The overall experimental unit setups for the interaction experiments are nicely designed and thoroughly described.

-It is unclear how "tending duration" was actually measured.

-I have some concerns over sample size, as the aphids on leaves and stems were divided across three different types of plants. Are the stems and leaves truly comparable across these plants in such a way that position on the leaves of sugarcane can be equally compared to position on the leaves of sorghum and Johnson grass?

-Some other questions/considerations that the authors may want to take in the study design, or at least address in the discussion:
1.) Would you expect tending frequency on leaves to be more comparable to tending frequency on stems if a plant with aphids on leaves did not have any aphids on the stem?
2.) Since distance from the ant nest and discoverability could also determine which group of aphids (on stems versus on leaves) are tended at the highest frequency, what if aphids on leaves were placed closer to the nest than aphids on stems?
3.) Would you expect there to be a saturation point of ants on aphids, such that once aphids closest to the base of the plant (on stems) have become over-tended by ants, tending might begin to increase on aphids more distant from the base (on leaves)?

Validity of the findings

-Studies have previously documented the increase of biomass of aphids and ants as a result of symbiosis, but this model system between TCA and sorghum aphids is poorly understood and unexplored. This study aims to fill that need, though I think the authors could really sell the importance of their results further.

-The conclusion appropriately restates the hypotheses and findings of the work, but new information is presented throughout that should have been presented either in the introduction or the discussion. The conclusion should focus more on wrapping up the manuscript with the broader significance of the work.

-I recommend including more supplemental information about collection locality and voucher specimens.

Additional comments

While the study system and the research presented are critical for understanding interactions between invasive species and their potential agricultural and ecological impacts, I think this manuscript is in need of major revision, largely due to concerns with sample size/study design. In particular, I am most concerned with how positions (leaves and stems) were grouped together across three different study plants. A more appropriate study design would include larger sample sizes for each plant. It appears as though they were grouped together in this way to improve the overall sample size for testing position, but I am not confident that this is biologically justifiable in practice.

Please see my line-by-line comments in the attached document for more details.

I wish the authors the best and look forward to seeing the final manuscript.

Annotated reviews are not available for download in order to protect the identity of reviewers who chose to remain anonymous.

---

## Round 0.2 · accepted · Accept

The corrections/concerns have been significantly affected by the authors, and the reviewers agreed with the correction and concerns. Point-to-point corrections were cross-checked and the manuscript is ready for publication.

Reviewer 1 ·

Basic reporting

After overviewing the revise version of MS, all the required flaws are included in the manuscript. Now results are quite good and well-written. Discussion is well written and is a reasonable analysis of the results.

Experimental design

Experimental design and lay out are well explained.

Validity of the findings

The finding are valid and will add to the current body of literature.

Additional comments

All the previously mention comments and short comes are incorporated in the manuscript so the decision is accepted

Reviewer 2 ·

Basic reporting

.

Experimental design

.

Validity of the findings

.

Additional comments

I looked over the responses to my feedback and the revised manuscript and am satisfied with the changes. My original concerns have been addressed.